# Analysis of Microstructure and Mechanical Properties of AlSi11 after Chip Recycling, Co-Extrusion, and Arc Welding

**DOI:** 10.3390/ma14113124

**Published:** 2021-06-07

**Authors:** Piotr Noga, Lechosław Tuz, Krzysztof Żaba, Adam Zwoliński

**Affiliations:** 1Faculty of Non-Ferrous Metals, AGH University of Science and Technology, A. Mickiewicza Av. 30, 30-059 Kraków, Poland; krzyzaba@agh.edu.pl (K.Ż.); adam.zwolinski@agh.edu.pl (A.Z.); 2Faculty of Metal Engineering and Industrial Computer Science, AGH University of Science and Technology, A. Mickiewicza Av. 30, 30-059 Kraków, Poland; ltuz@agh.edu.pl

**Keywords:** arc welding, recycling, aluminum alloys, mechanical properties, microstructure

## Abstract

Recycling of raw materials and is crucial for the production of new products for the global economy. The aim here is, on the one hand, to reduce energy consumption, and, on the other hand, to obtain materials with similar functional properties. The study undertook research on the possibility of processing AlSi11 aluminum chips by compaction and co-extruding to obtain a product in the form of a flat bar with mechanical properties not lower than those of the cast materials. The performed tests and the developed technique allowed to obtain flat bars with more favorable mechanical properties (Yield Strength YS ≥ 155 MPa; Ultimate Tensile Strength UTS ≥ 212 MPa) than the castings (YS ≥ 70 MPa ≥ 150 MPa). The weldability evaluation tests revealed that the material is susceptible to porosity. The presence of pores, which reduces the cross-section (up to 60%), reduces the tensile strength (up to 20 MPa). The typical joint structure and plasticity is obtained, which indicate the possibility of tensile strength improvement.

## 1. Introduction

Sustainable economic development constantly puts emphasis on searching for methods of manufacturing products with minimal energy consumption and minimal input of materials from primary sources. A highly effective way to achieve these goals is to close the material cycle in the economy by recycling waste materials and scrap. It is of particular importance in the case of light metals and their alloys, which due to their properties constitute an important material group in the global economy. Currently, the world production of primary aluminum is around 50 million tons per year, and the demand for aluminum has been steadily increasing in recent years at the rate of 2.4% per annum [1].

However, it should be taken into account that aluminum alloys are the most energy-consuming raw materials already at the production stage in the initial extraction processes from primary sources [2]. The most effective way to save both energy and primary sources resources seems to be closing the cycle of light metal materials through their recovery. It is estimated that recycling 1 ton of aluminum saves 6 tons of bauxite and prevents the emission of 9 tons of CO_2_ [3]. It is assumed that 1 kg of produced aluminum is responsible for the emission of 4 kg of CO_2_. [4] On a global scale, aluminum recycling prevents the emission of over 100 million tons of CO_2_ each year. Hence, there is a lot of research and experimental work related to the simulation of aluminum production and recycling and models allowing for the reduction of CO_2_ emissions [5], increasing the efficiency of aluminum-recycling processes by cryolite addition to the liquid metal [6] or preliminary scrap separation [7].

Almost all modern aluminum recycling is based on the processes of melting scrap. Various types of melting furnaces are used for this purpose, from gas furnaces, which are relatively cheap, to technologically advanced electric furnaces, which are equipped with special charge-introduction systems into the liquid metal [8]. In the case of large-size scrap and, in particular, aluminum scrap from food cans, recycling is widely used, mainly due to the universality and dimensions of this waste. On the other hand, the management of small and contaminated aluminum scrap remains a technological challenge, and material losses based on the traditional melting method may reach even 50% in some cases compared to the initial charge [9]. Such a low efficiency of the recovery of the fragmented material fractions is due to the high chemical activity of aluminum and its alloys and the process of destroying passivation barriers at high temperature during heating to the melting point. Hence the possibility of recycling small fractions of metals and their alloys, is limited. Small fractions of metals pass almost entirely to the skimming and constitute a very hard and heavy post-process waste. 

In recent years, there has been a lot of research work on the recycling of aluminum alloy chips using modern and innovative techniques without the liquid phase [10,11,12,13,14,15,16,17,18,19,20,21,22,23,24,25]. 

To reduce a losses can be the recycling carried out at a temperature much lower than the melting point. At low temperatures the intensive oxidation is not observed which allow the direct processing of the fragmented fractions of scrap [10]. This possibility is offered by the hot extrusion technology. This method uses the formation of an unoxidized, newly formed surface resulting from a strong plastic deformation. The newly formed surface may have the ability to form cohesive bonds which lead to the fusion of the dispersed metal fractions into a solid form. The extrusion allows to obtain a solid material from various types of starting materials, such as powders, where the strong cohesion of fine particles [11] without voids and porosity [12] is observed. 

Such materials can be mechanically synthesized and strengthened by the addition of metal oxides [13,14] or subjected to special processes, such as rapid solidification (RS). These processes allow to obtain better mechanical properties of materials compared to the conventional methods [15]. 

Gronostajski J.Z. et al. [16] carried out research on the consolidation of dispersed aluminum alloys in the form of flakes using the hot extrusion process. Due to the use of a low extrusion speed, the good quality profiles with a density close to that of the solid material was obtained. Misiołek W. et al. [17] investigated the effect of pressing the initial charge in the form of Al 6060 alloy flakes, and the selection of the deformation path for the final results of mechanical properties through the use of the equal channel angular pressing (ECAP) process. The proposed technology gives very promising results in terms of energy savings and the production of high-quality aluminum structural profiles from dispersed metal forms. Wzorek Ł. et al. [18] carried out the consolidation tests and compared the results of mechanical and physical tests of the 4XXX series alloy in the form of flakes, casts, and strips. To find the best material, the quality level of each tested materials was assessed using the quality ratings proposed by R. Kolman. The research showed that the material with the best mechanical properties is obtained from chips after the milling process, produced by plastic consolidation.

Koch A. et al. [19] used aluminum chips pre-pressed, sintered field-assisted sintering technology (FAST), and then extruded without the solid phase, which allowed for obtaining higher fatigue properties than in conventional recycling methods. An innovative method was presented by M. E. Mehtedi [20], in which he combined co-extrusion with friction mixing, ensuring a high-quality outer surface, while inside there were various sizes of porosity. On the other hand, sintering was used for the recycling of AA6061 aluminum chips where by applying pressure the aluminum chips are combined with a binder such as zinc stearate C_36_H_70_O_4_Zn [21]. 

In the cyclic extrusion compression back pressure (CECBP) technique fine chips of AA 6061 alloy were pre-compacted under a high pressure and then subjected to the forging process, which allowed high compaction of the fragmented material, resulting in similar hardness as in the case of solid material [22]. Baffari [23] proposed an innovative solution by applying the friction stir consolidation (FSC), where the AA1050 chip waste materials were put into a chamber with a rotating pin, which caused the material to soften and to consolidate finally. The material subjected to this process has similar mechanical properties as the materials obtained from conventional recycling. Another innovative approach to recycling AA5083 chips was proposed by J.B. Jordon [24], using the additive frictional mixing deposition technique, which allowed to avoid problems related to the presence of oxides, as is the case with conventional recycling. B. Li [25] dealt with the recycling of Mg–Gd–Y–Zn–Zr alloy chips in a solid state, using spark plasma sintering (SPS). The research revealed that this process gives a very high recycling efficiency, and the post-recycling material obtains properties similar to the solid material.

In summary, a review of previous studies shows that although the properties and structure of materials after consolidation are well-known, there are no reports on the possibility of joining them by welding, which results from the limited weldability of the casting alloys due to the high porosity of the material after casting [26,27]. Hence, taking the above issues into account, this paper focuses on the weldability of the recycled AlSi11 alloy bars obtained by ingot machining (turning and milling) to reach two types of chips with different size and morphology, and then two-stage consolidation and hot co-extrusion. The choice of the type of aluminum alloy was made due to the widespread use of this material in the various industries, such as automotive or shipbuilding [28,29]. AlSi11 alloy is characterized by high electrical and thermal conductivity, good machinability, crack resistance, low casting shrinkage, and high castability. These properties are mainly related to the presence of silicon, which increases the strength of the alloy while reducing its plasticity [30]. The flat bars obtained by hot co-extrusion and its welded joints were subjected to visual tests, mechanical properties (Ultimate Tensile Strength UTS, Yield Strength YS, Elongation A, Vickers Hardness HV2), fractographic tests, measurements of density and porosity, chemical composition, and microstructure. 

The paper presents the results of the tests carried out for the recycled AlSi11 alloy, made to determine the influence of the chip size on mechanical properties of extruded bars and its weldability. 

## 2. Experimental Methodology

### 2.1. Materials for Tests

The material for the tests was eutectic aluminum alloy of AlSi11 grade in the form of a commercial ingot with dimensions of 70 mm × 100 mm × 700 mm. The chemical composition and mechanical properties of ingots and filer metal for welding are presented in Table 1 and Table 2, respectively.

### 2.2. Bars and Joints Preparation

The samples manufacturing plan is shown in Figure 1. Two types of chips with different size morphologies were produced from the AlSi11 ingot (reference material) by machining (turning and milling). Machining was carried out in laboratory conditions with the cleanliness of the stand and the absence of coolant in the process. The first material was fine chips with a fraction of 0.16–0.4 mm (Figure 2A) produced on a ROBGRAF 1 milling machine (Markus-Texi, Ostrowiec Św., Poland), where the cutter moved in a spiral motion, starting from the center and ending with the outer part of the charge with a diameter of 40 mm. The milling process was carried out at a depth of cut of 0.5 mm/tooth (mm/rev) and a rotational speed of 28,000 rpm. The second type of chips was the coarse spiral chips with average dimensions: 22 mm × 4 mm × 0.5 mm (Figure 2B), produced in the turning process at the TUM 35 station (Famot, Pleszew, Poland) with a rotational speed of 315 rpm, with a feed rate of 0.2 mm/s. Average size of the chips was measured by the planimetric method using the ImageJ software version 1.53j (Rockville, Bethesda, MD, USA).

The reference material, with which the two types of chips were compared, was a solid ingot cast by gravity into a steel crucible with a diameter of 40 mm and a height of 60 mm (Figure 3A). The dimensions of the crucible correspond to those of the recipient of the hot extrusion press used further in the experiment. To obtain the flat bars for welding, the obtained chips were subjected to a two-stage process, including:

1. Cold compaction (Figure 3A). Molded pieces were produced from clean chips (30 g), put into a die with a diameter of 40 mm, and subjected to a pressure of 100 MPa using the PS Logistics 100 press (Racot, Kościan, Poland) (Figure 4B,C). 

2. Hot co-extrusion (Figure 3B). The charge in the form of six compacts was placed in a recipient heated to 375 °C. The moldings were annealed for 20 min and then subjected to hot co-extrusion at a speed of 3.7 mm/s using a die with a cross-section of 3 × 15 mm on a hydraulic press (Zakład Mechaniczny Hydromet, Bytom, Poland), obtaining flat bars intended for welding.

The material prepared in the form of flat bars was cut into sections of 150-mm long, then V-beveled, maintaining the bevel angle of 30° (groove angle 60°) and the threshold of 1 mm. The distance between the joined elements was 1 mm (Figure 5). Before welding, the material was mechanically cleaned to obtain a metallic surface, degreased and thoroughly dried. Butt-welded joints were produced by the metal inert gas (MIG) method using the RobSpaw welding robot (iPowerInstall, Juszczyna, Poland) and the welding power source MIG 280 DUAL-PULS Syneria (Magnum, Kielce, Poland). The welding process was carried out on the basis of a synergic double pulse line with a maximum pulse voltage of 20 V and duration as well as a base voltage of 19 V and a current intensity in the range of 100–110 A. The welding speed was 12.5 mm/s. The heat input was 0.138 kJ/mm. In the welding process, AlSi12 welding wire with a diameter of 1 mm was used. Argon (99.998% Ar) was used to shield the face and root of the weld.

Samples of the base material (bars) and welded joints used in the tests were determined in accordance with Table 3.

### 2.3. Research Scope

The extruded flat bars (profiles), intended for welding, were tested in terms of quality assessment and meeting the requirements of EN 1706 [31] covering basic mechanical properties, i.e., tensile strength, yield point, elongation and hardness, and additionally measurements of density and microscopic observations in terms of structure and surface morphology. The diagram of testing range and scope of bars after extrusion and welded joints is shown in Figure 6.

The joints were visually inspected in order to identify the possible welding defects. Visual examinations were carried out maintaining a direct path between the observer and the sample. The lighting during the test was 550 lx. Samples were taken from the joints for metallographic and mechanical properties tests using the BP05d electro-erosion machine (Zapbh, Końskie, Poland). The samples for the microstructure tests were mounted in Bakelite with carbon filler resin on the CITOPRESS-1 device (Struers, Copenhaga, Denmark). The metallographic specimens were ground on water-based abrasive papers and polished with the use of diamond pastes with a gradation up to 1 μm and the OPS polishing agent.

Metallographic specimens were used in macroscopic, microscopic, and SEM examinations. Macroscopic observations of the joints were carried out using the Leica S9 stereoscopic microscope (Leica Microsystems, Heerbrugg, Switzerland), while the observations of the joints microstructure were made using light microscopy (LM) using Leica DM/LM (Leica, Wetzlar, Germany) and scanning electron microscopy (SEM) using Phenom XL (Thermo Fisher Scientific, Waltham, Massachusetts MA, USA). SEM observations were carried out at a beam-accelerating voltage of 20 kV. Chemical composition analyzes were performed using SEM-EDS (Hitachi SU-70, Hitachi Ltd., Tokyo, Japan). 

The mechanical properties in the static tensile test were made with a Zwick Roell Z050 testing machine (Zwick/Roell Group, Ulm, Germany) on samples with base dimensions of 15 mm × 5 mm. The tension speed was 8∙10^3^·s^−1^. The obtained fractures were subjected to macroscopic observations using the Stemi 305 stereoscopic microscope (Zeiss, Oberkochen, Germany) and the scanning electron microscope (SEM, HITACHI SU70, Tokyo, Japan) used in metallographic research.

Density was measured using the displacement volume method using a laboratory balance XA 120/250.4Y (Radwag, Radom, Poland). Measurements were made on samples with dimensions of 15 mm × 40 mm with an measurement accuracy of 0.001 g. 

The porosity of the material was analyzed on the basis of macroscopic photographs using Sigma Scan Pro software (Version 5, 2007, Systat Software, San Jose, CA, USA).

## 3. Test Results 

### 3.1. Base Material-Extruded Profiles

The surface morphology of the flat bars obtained by hot co-extrusion is shown in Figure 7. All the samples produced have a similar surface morphology with no visible cracks. On the other hand, there are visible small surface defects (up to approx. 1–2 μm) like local indentation or material losses.

Measurements of the density of the materials did not reveal any significant differences between the different processing states (Table 4), where the difference in density is less than 0.006 g/cm^3^. This indicates that the obtained results are in the range of the measurement error.

The microscopic observations (LM/SEM) revealed a similar structure in all the tested materials. The reference material (M1) is characterized by relatively large areas of α-Al phase, against which there are visible precipitations in the form of massive and lamellar form corresponding to the primary silicon crystals and brittle intermetallic phases (Figure 8). In the case of M2 and M3 materials, the fragmentation of the structure in relation to M1 is observed (Figure 8). For M1 material, the mean diameter of precipitation is 3.05 μm, and for samples M2 and M3, 1.45 μm and 0.9 μm, respectively. The separations of Si and its compounds from Fe and Cr are evenly distributed and much finer than in the case of M1 (Figure 9). This indicates that the starting material, as a result of machining (turning/milling) and hot co-extrusion, was crushed without changing the chemical composition of the precipitates. Observations of the surface in the cross-section revealed porosity in the area of material M1 and M2 of approx. 3% (Figure 8A,B), while strip discontinuities are additionally observed in M3 (Figure 8C). 

Hardness measurements revealed that M3 material is characterized by the highest hardness amounting to approx. 75 HV2, which is approx. 15 HV2 higher than the original material, while for material M2 the hardness recorded is approx. 65 HV2 (Table 5). In addition, both the M2 and M3 material exhibit higher strength than the original M1 material. In both cases, the yield point was approximately 25% higher than for M1 (Table 6). The stress–strain curves are shown in Figure 10. The shape of curves for M2 and M3 is similar to M1. Fractographic observations revealed a ductile fracture for all tested materials (Figure 11).

The obtained results of tests of materials in the state after hot extrusion show that the reference material M1 is characterized by a relatively low porosity and a fine-grained structure. Porosity occurs both in the reference material and obtained from chips, and has a similar morphology and is evenly distributed over the cross-section. Profiles made of fine (M3) and large chips (M2) have a lower porosity than casting material (M1), while in the M2, a greater number of large precipitates and the presence of impurities bands associated with extrusion are observed. For samples obtained from large chips (M2), delamination in the form of strips arranged parallel to the material flow during extrusion was observed. 

The materials obtained from chip recycling have a fine-grained structure and more favorable strength properties than the M1 material (Figure 9, Table 7). The recycled chip material has higher mechanical properties than the remelted and extruded material, while meeting the requirements of EN 1706 for AlSi11 alloy (EN AC 44000). The presence of these defects and their directionality may indicate a strong anisotropy of the mechanical properties, especially in the direction of the material thickness. The small thickness of the material makes it impossible to perform tests in this direction.

### 3.2. Welded Joints

Visual tests showed the regular shape of the welds made on all flat bars, regardless of the method of their production. This indicates that the welding parameters used should ensure high-quality joints. In the case of samples W2 and W3 made of flat bars corresponding to materials M2 and M3, an increase in the height of the excess weld metal in relation to M1 was noted and no visible scaling of the weld in both joints was observed. Detailed observation of surface revealed a typical morphology of the weld face W2 with very small pores (see Figure 12A,B and Figure 13B). In addition, a face porosity was disclosed for the W3 joint (Figure 12C).

Macroscopic observations in the cross-section of the joints (Figure 13) revealed their porosity, while in the joint W1 the pores are relatively small and evenly distributed, where the total amount of pores in the cross-section of the joint is 5.2 ± 0.3%. Uniform distribution of pores is also observed in sample W3, where their number and size cause that they cover most of the cross-section of the weld (60 ± 2.1%). The largest pores are in the excess weld metal area (Figure 13C). In the W2 joint, the porosity is mainly located in the area of the melted edges of the base material (Figure 13B). The total amount of porosity in the W2 joint is 27 ± 1.2%. Observations of porosity using SEM reveal their regular, globular shape for W1 and W2 (Figure 14A,C and Figure 15A,C), but also an irregular shape for W3 (Figure 16D), showing clusters of porosity and tubular blisters. In the W2 and W3 joints, near the fusion line, small gas voids (less than 50 μm in diameter—yellow arrow) are observed, while in the middle part—much larger (200–500 μm in diameter—yellow arrow). The results of the porosity measurement in relation to the joint cross-sectional area revealed an increase in porosity (Table 7). The obtained results of the porosity caused an increase in the cross-sectional area of the weld and height of the excess weld metal of W2 and W3. Macro- and microscopic examinations did not reveal a change of the structure morphology in the heat-affected zone (HAZ) for all joints—Figure 13, Figure 14C, Figure 15C and Figure 16C. 

Microscopic examinations carried out with the use of LM and SEM revealed a change in the morphology of silicon precipitates and intermetallic phases in all welds (Figure 14D, Figure 15D and Figure 16D) in relation to the base metal (Figure 14A, Figure 15A and Figure 16A). The welds in W1, W2, and W3 are characterized by a dendritic structure with the presence of eutectics in the inter-dendritic areas (Figure 14D, Figure 15D and Figure 16D). The presence of the Al-Si eutectic (presence of the β-Si phase in the form of lamellae) was revealed in the interdendritic regions. In addition, the analysis of the element concentrations revealed the occurrence of segregation of alloying elements in the interdendritic regions, mainly Fe, Mn, and Cr. The precipitations also take the form of plates and needles (Figure 14E, Figure 15E and Figure 16E). The smallest dendrite cores were revealed in the W1 joint. The largest cores were observed in the W2 and W3 joints.

The results of testing the mechanical properties of welded joints are shown in Figure 17 and Table 8. The tests revealed that the joint W1 has a higher UTS (173 MPa) than the base material M1 (167 MPa), and the rupture occurred in the area outside the weld. This indicates that during welding, heat treatment (partial aging) occurred due to the heat input into the material. The obtained UTS in W2 (60 MPa) and W3 (20 MPa) is much lower than M2 (212 MPa) and M3 (224 MPa), respectively, and the fracture occurred in the area of the weld (see Table 6 and Table 8).

All fractures are characterized by a quasi-plastic fracture. In the case of W1 the material is deformed, but cracking was due to the presence of brittle Si-plates precipitation and intermetallic phases (Figure 18A). The low strength in W2 and W3 (Table 8) results from a significant reduction in the cross-section of the weld, with 27% and 60% porosity, respectively (Table 7). The material between the pores is characterized by a quasi-ductile fracture typical of Al-Si alloys (Figure 18B,C). Observation of the inside surface of the porosity indicates an insufficient amount of liquid metal and the possibility of a hot crack (Figure 18C–yellow arrow, Figure 19). The large number of pores and their smooth surface indicate that during welding in the molten metal (weld-pool) there was a large amount of dissolved gases—e.g., hydrogen, which caused porosity during crystallization.

Hardness distributions (HV2) in the cross-section of the welded joint W1 revealed an increase in hardness compared to the base material by approx. 15 HV2 (Figure 20A). In the case of W2 and W3 welds, a decrease in hardness to 38 HV2 for W2 and 25 HV2 for W3 was observed (Figure 20B,C). Low hardness values result from a significant amount of porosity in the weld area (Table 7, Figure 13).

### 3.3. Results Discussion

The test results show that while it is possible to produce a material in the form of a flat bar or a strip of recycled chips with mechanical and plastic properties meeting the requirements for castings, these materials show a high tendency to porosity and lower mechanical properties during welding.

The tests of materials after hot co-extruding have shown that, regardless of the size of the chips used, it is possible to obtain good mechanical properties, i.e., YS > 120 MPa, UTS = 167 Mpa, and A > 11%. The obtained results are consistent with the values [32]. This shows that the recycling method used for production allows for obtaining a full-value base material. The produced materials are characterized by low porosity (approx. 3%), and the pores assume a globular shape across the cross-section to the rolling direction. As is well-known, gases and other contaminants on the chip surface and from the cutting process can be trapped inside the pores.

MIG arc welding of flat bars revealed the presence of significant porosity of the welds, and the gas voids cover a significant part of the cross-section (up to 60% for W3)—Table 7. The porosity results from the presence of gases emitted from the base material, which is indicated by the macroscopic tests performed and the location of large number of pores and the precipitation of iron and manganese-rich intermetallic phases near the fusion line. This shows that without refining the liquid metal and degassing the metal bath, it is not possible to obtain high weldability, and the technological issues will not allow to avoid this porosity of the welds. In the welding process, when the edges melt, gases are introduced into the metal bath (weld pool) and they dissolve (no degassing occurs). At the same time, heating the material to a temperature above 230 °C (in HAZ area) causes the diffusion of hydrogen to the weld pool, and if it goes to the metal bath, it also dissolves. During the cooling of the weld pool, when crystallization begins, a layer of solid metal forms on the surface of the pool, preventing the degassing of the metal, and at the same time the dendrites from the fusion line cause the gases to breakdown and finally their progressive diffusion and trapping in the gas voids. This phenomenon can be observed on the sample of material M2 and the weld W2 (Figure 13B), where in the area of the weld near the fusion line, the gas voids are densely arranged, but they are small. The observed macrostructure indicates a source of gas is the base material. The observed porosity of the W3 weld results from the size of the used chips and additionally the presence of inclusion bands rich in Fe. In the areas of the bands may also be a gas, which can create significant gas voids (up to 500 μm), and at the same time, high gas pressure causes some to be pushed out of the pool and then, during crystallization, on the surface they have the character of open pores (Figure 12C).

The gases were found in the base material, and as a result of heating to high temperature and melting the edges, they escaped into the liquid metal. The gas favoring the formation of porosity is hydrogen, the solubility of which in liquid aluminum increases with increasing temperature, where at 660 °C it is 0.69 cm^3^/100 g of the weld metal, and at the temperature slightly below the melting point (solidus) it decreases to a value below 0.036 cm^3^/100 g of weld metal [33]. As a result, the dissolved gas is removed from the metal during crystallization. The dendritic nature of crystallization causes the gas to be trapped between the arms of the dendrites, creating porosity at the same time.

The study of mechanical properties shows that the strength of the obtained joints is much lower (over 60%) than that of the base material (Table 8). Here, the strength is significantly influenced by the porosity of the welded joint (Table 7). The base material (M1, M2 and M3) shows high tensile strength up to 167 MPa (M1) with relatively high elongation (18%). The effect of the heat input into the material resulted in a decrease in plasticity (6%) and a slight increase in the material strength (173 MPa). In the case of W2 and W3 joints, the obtained mechanical properties are very low and do not exceed 20 MPa and 60 MPa, respectively. Low tensile strength and also low hardness in the weld area are the result of their high porosity (27% and 60%, respectively) and a reduced cross-section. At the same time, the size of the pores shows that while the small pores have a slight influence on the mechanical properties (W1), the large and irregularly shaped pores are already the area where damage occurs at low loads (W2 and W3). Fractographic studies showed that the fracture ran along the plane of large pores, i.e., the smallest cross-section of the joint (Figure 18). On the other hand, the fracture is quasi-ductile, with visible traces of plastic deformation, and the β-phase plates are visible on the fracture surface. The analysis of the alloying elements concentrations showed that these precipitates are rich in silicon (α+Si eutectic, β-Si phase) and other elements, mainly Fe and Mn (e.g., AlSiFeMn intermetallic phase) [32]. The secretions are small and located in the interdendritic regions.

## 4. Conclusions

The tests revealed that it is possible to obtain materials by recycling, which will have a favorable structure with lamellar and acicular precipitates of silicon and intermetallic phases, and also favorable mechanical properties meeting the requirements for eutectic castings such as AlSi11 alloy. The degree of porosity (up to 60%) obtained by compaction and extrusion does not differ from the porosity observed for cast alloys, which is indicated by similar density results (2.66–2.67 g/cm^3^). 

The presence of porosity as well as iron-rich intermetallic phases shows that these materials will be characterized by limited weldability resulting from the extraction of gases from the base material. Depending on the type of chips used in the production of the material, the porosity of the weld seam varies—for coarse chips (W2) it is approx. 27%, and for fine chips (W3) approx. 60%. However, the degree of porosity does not clearly affect the mechanical properties of the joints, where the lower porosity (27%) and higher hardness 38 HV2 in W2 do not prevent the formation of the small cross-sectional area being the shear plane in the static tensile test (UTS 60 MPa). Larger pores, but evenly distributed in the cross-section of the welds (W3), have no effect on higher tensile strength (UTS 20 MPa), behaving similar to metal foams during tension [34]. 

Summing up, the obtained results show that while it is possible to recycle casting aluminum alloy chips and obtain, by extruding, full-value materials, which may also be economically feasible [35], further work is required in the field of weldability, especially to reduce the weld porosity. It should be assumed that limiting the amount of heat input into the material or joining without a liquid phase (without melting) should result in satisfactory properties, for example FSW [36].

Overall, the research showed that: The size of the chips used in the recycling process affects the mechanical properties of the materials after co-extrusion, where more favorable properties are obtained for thick chips (UTS upto.224 MPa);Chip size affects the porosity of welded joints, where lower porosity (27%) is obtained for thick chips, while the distribution of pores (gas voids) in the cross-section is uneven, and greater (60%) for fine chips, which results in less favorable mechanical properties (UTS 20 MPa, YS 15 MPa).The type of chips used did not affect the structure of the weld and the character of the fracture, and the increase in porosity was reflected in the height of the excess weld metal;The tests performed revealed the possibility of obtaining metallic continuity of the AlSi11 alloy, the equally low quality of the obtained joints indicates that joining tests should be performed using other welding methods with a limited amount of heat input into the material, i.e., friction stir welding (FSW).

## Figures and Tables

**Figure 1 materials-14-03124-f001:**
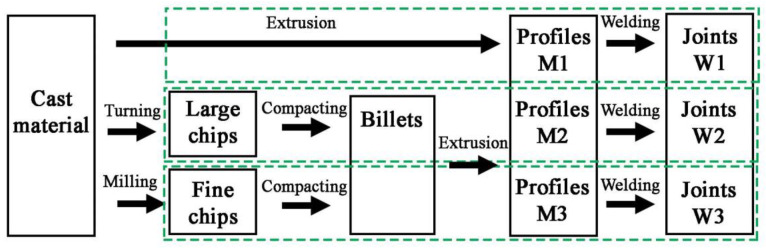
Manufacturing process of materials for tests.

**Figure 2 materials-14-03124-f002:**
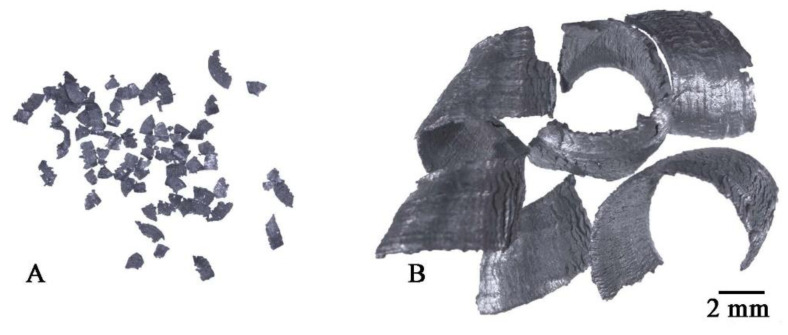
Chips after machining: (**A**) milling, (**B**) turning.

**Figure 3 materials-14-03124-f003:**
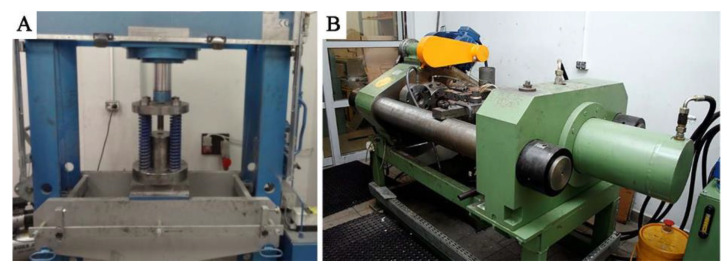
Laboratory installation for extrusion: (**A**) cold compaction, (**B**) hot co-extrusion.

**Figure 4 materials-14-03124-f004:**
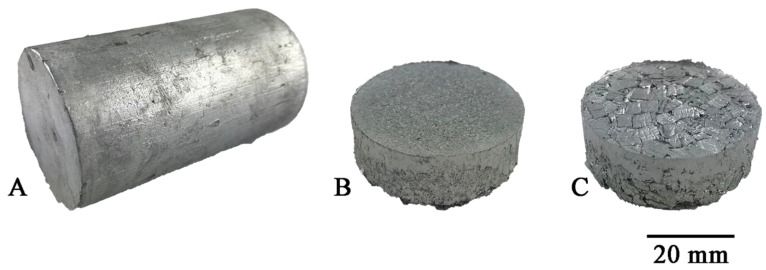
Recycled material for tests after: (**A**) casting, (**B**) fine chips consolidation, (**C**) large chips consolidation.

**Figure 5 materials-14-03124-f005:**
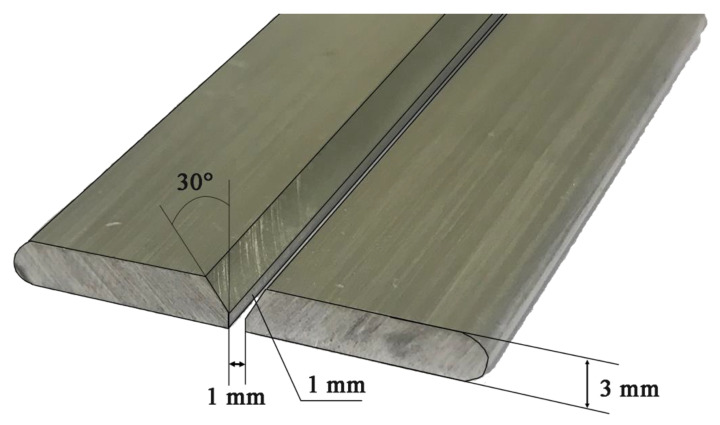
Plates beveling for welding.

**Figure 6 materials-14-03124-f006:**
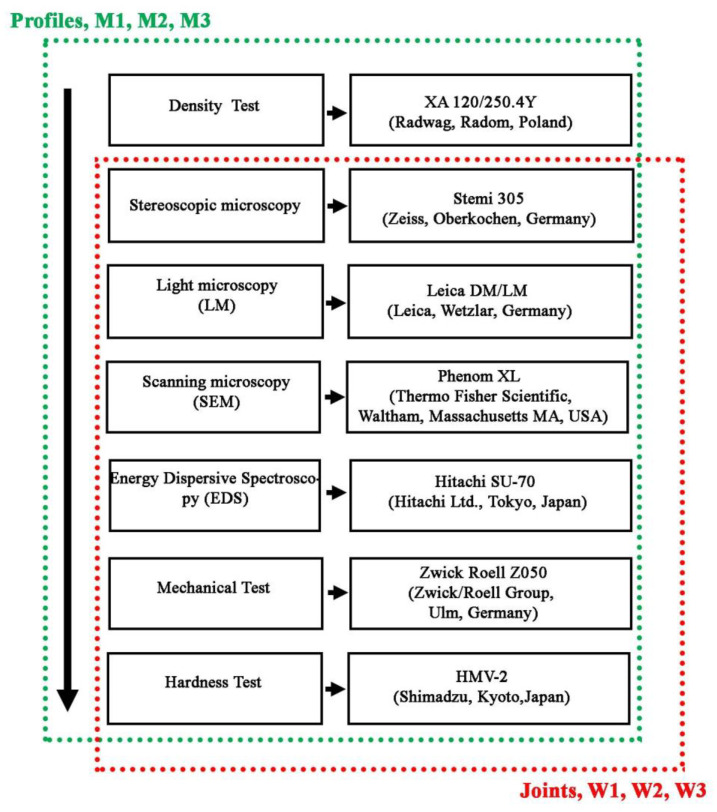
Test range for profiles after extrusion (green dot line) and welded joints (red dot line).

**Figure 7 materials-14-03124-f007:**
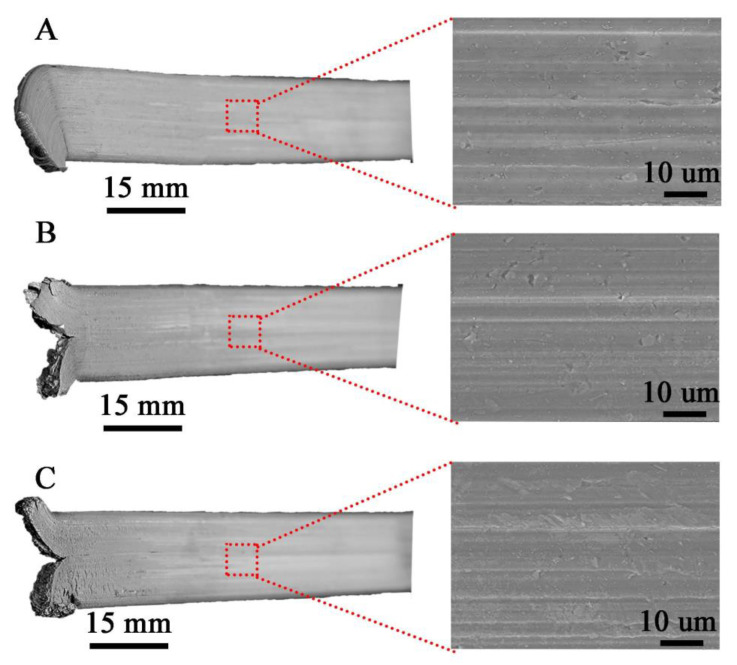
Flat bars after hot co-extrusion from: (**A**)—cast (M1), (**B**)—large chips (M2); (**C**)—fine chips (M3); surface morphology with lack of cracks.

**Figure 8 materials-14-03124-f008:**
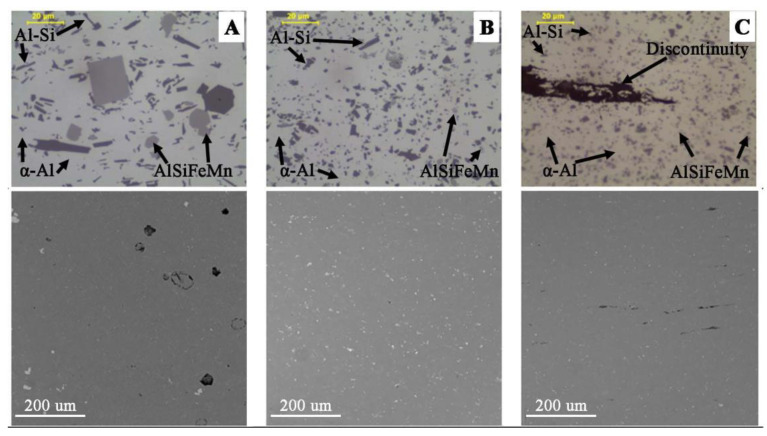
Microstructure in cross-section of extruded bars: (**A**) M1; (**B**) M2; (**C**) M3; visible main phases in microstructure (LM) and surface morphology in cross-section (SEM) with porosity of cast metal (**A**) and discontinuities of fine chips bar (**C**).

**Figure 9 materials-14-03124-f009:**
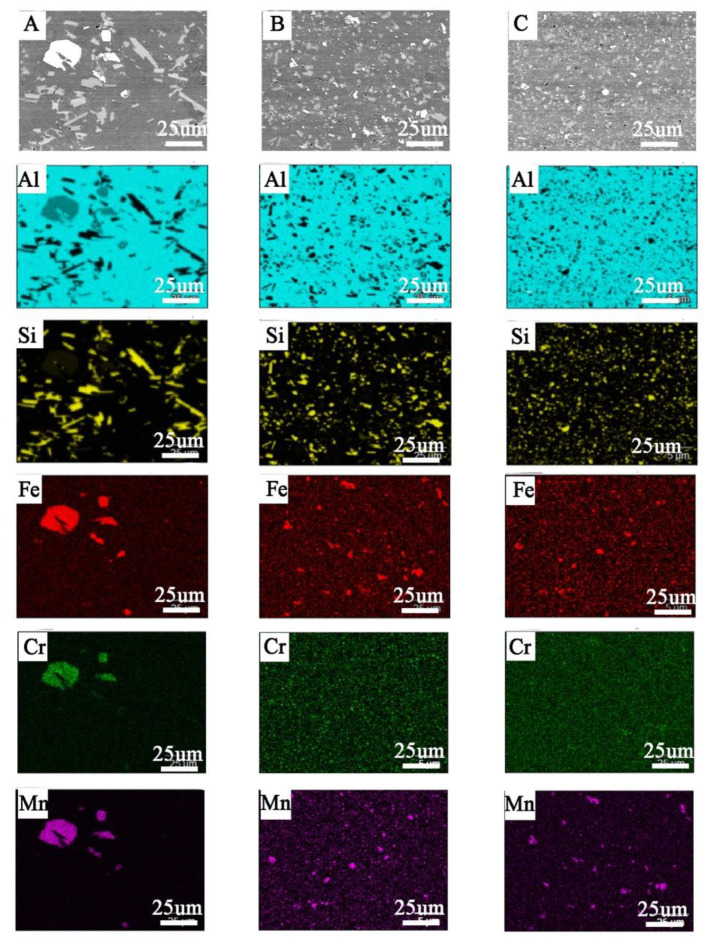
Elements distribution in cross-section of bars made by: (**A**) cast (M1), (**B**) large chips (M2), (**C**) fine chips (M3); visible refinement of structure without chemical composition changes.

**Figure 10 materials-14-03124-f010:**
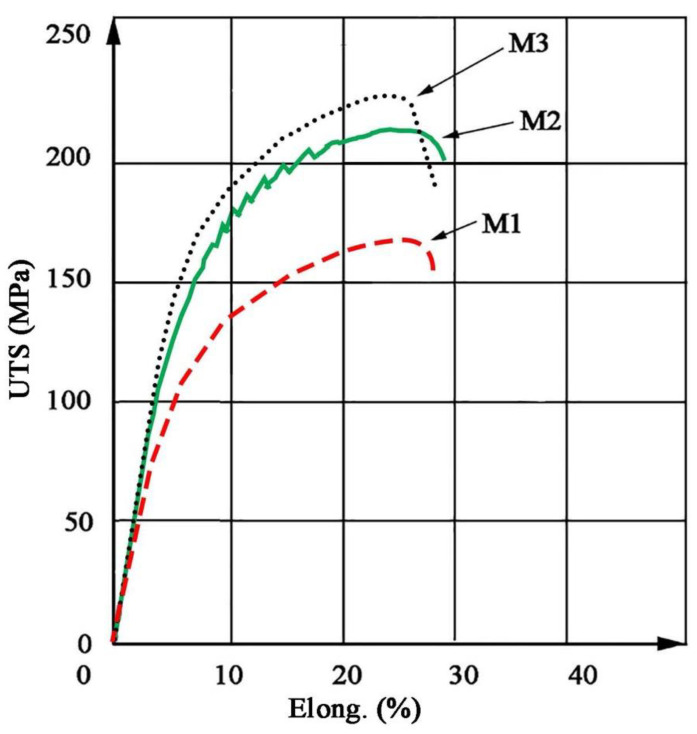
Stress–strain (elongation) curves for extruded bars.

**Figure 11 materials-14-03124-f011:**
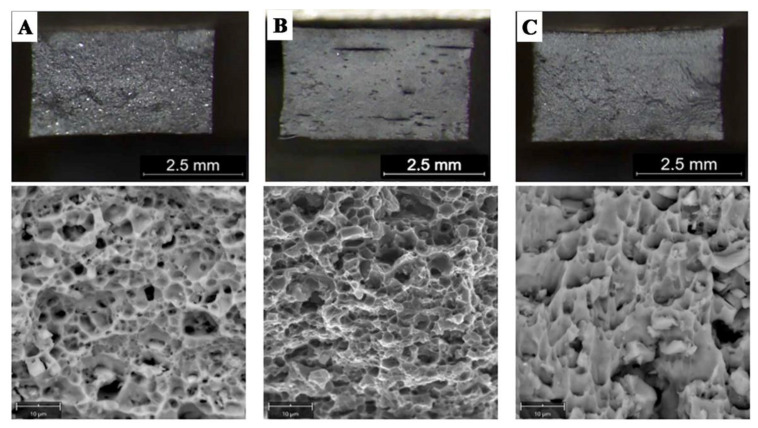
Fractures after tensile test (LM/SEM) of: (**A**) M1; (**B**) M2, (**C**) M3; ductile fracture in all samples.

**Figure 12 materials-14-03124-f012:**
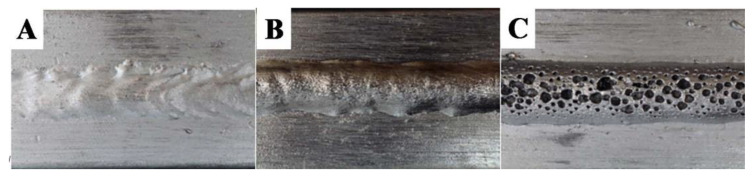
Face of welded joints:, (**A**) W1—correct shape, (**B**) W2—fine pores, (**C**) W3—strong porosity.

**Figure 13 materials-14-03124-f013:**
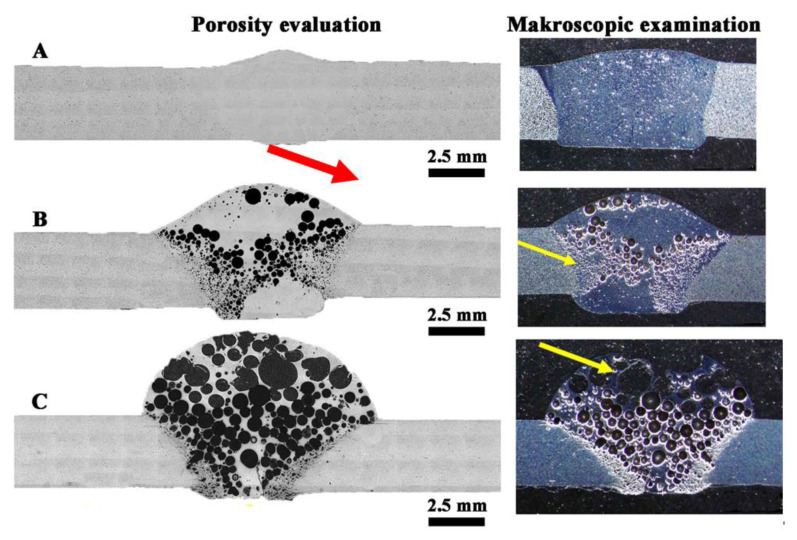
Macrostructure of welded joints: (**A**) W1, (**B**) W2, (**C**) W3; yellow arrows indicate the porosity; red arrow indicate small porosity visible on the face of the weld.

**Figure 14 materials-14-03124-f014:**
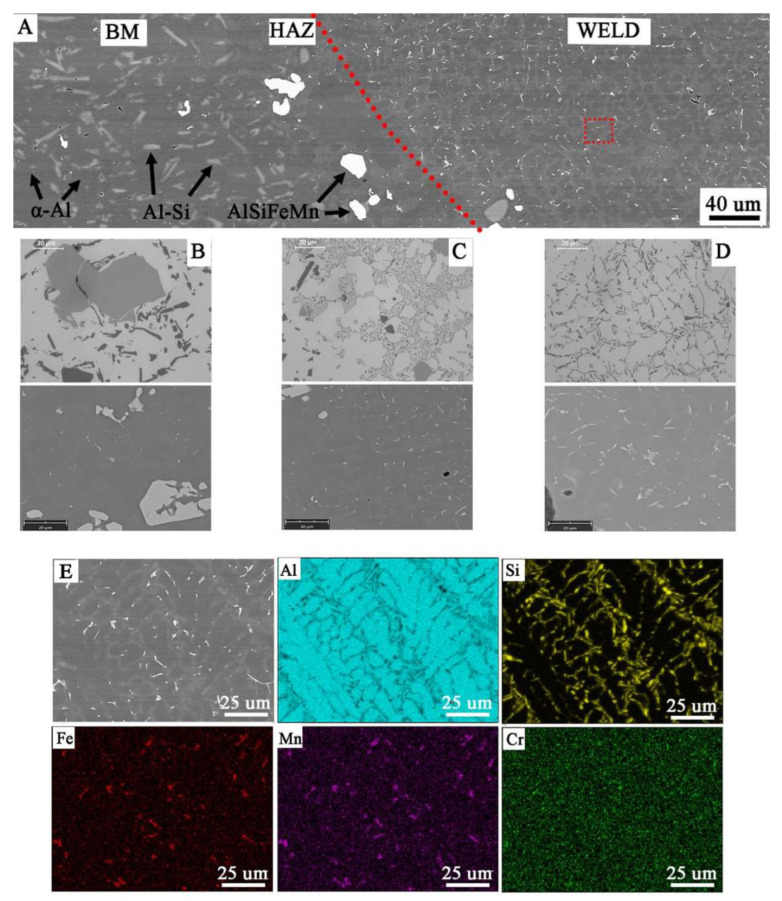
Microstructure of joint W1: (**A**) general view, microstructure of: (**B**) BM with coarse particles, (**C**) HAZ with refined particles near to fusion line, (**D**) weld with dendritic structure and eutectics in the inter-dendritic areas, (**E**) analysis of distribution of the most important elements in the weld metal (red rectangular indicate the EDS placement).

**Figure 15 materials-14-03124-f015:**
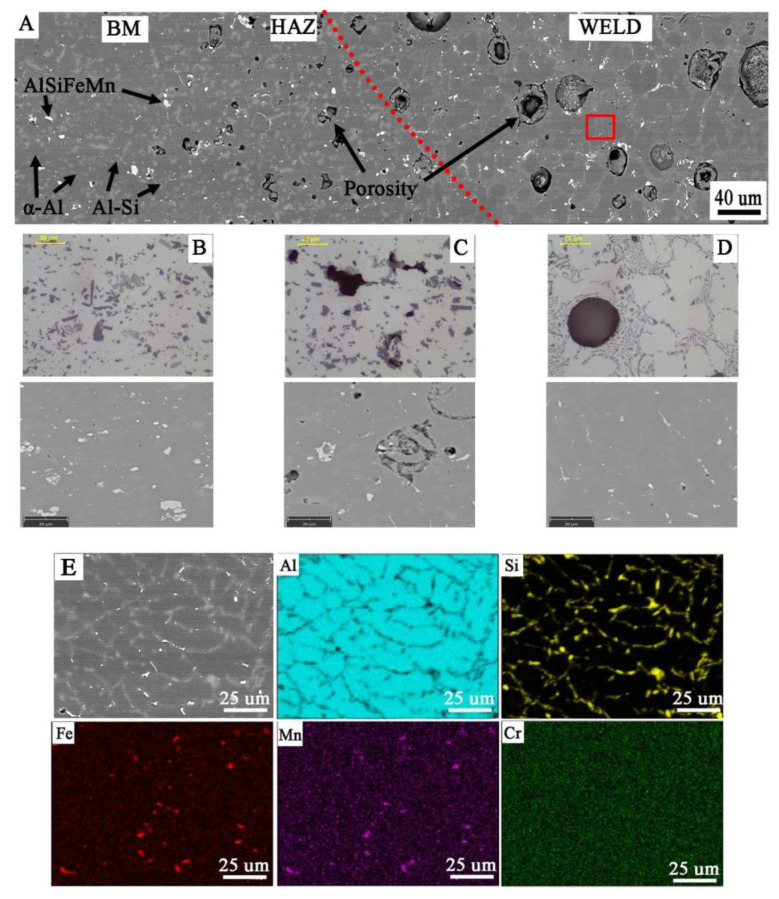
Microstructure of joint W2: (**A**) general view, microstructure of: (**B**) BM with coarse particles; (**C**) HAZ with refined particles near to fusion line; (**D**) weld with dendritic structure and eutectics in the inter-dendritic areas; visible porosity in HAZ and weld; (**E**) analysis of distribution of the most important elements in the weld metal (red rectangular indicate the EDS placement).

**Figure 16 materials-14-03124-f016:**
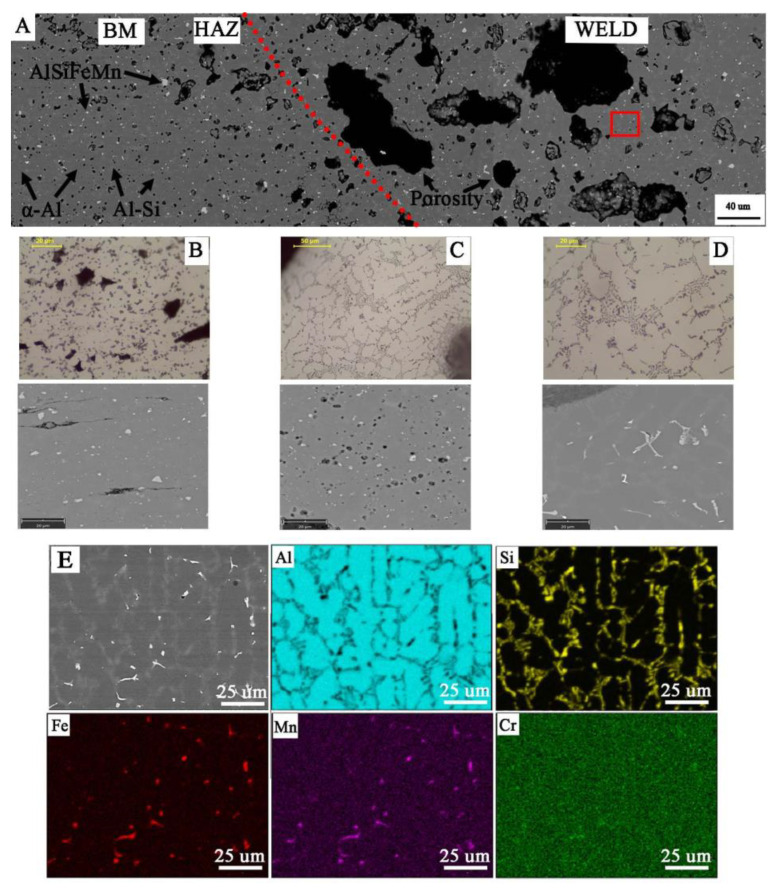
Microstructure of joint W3: (**A**) general view; microstructure of: (**B**) BM with coarse particles; (**C**) HAZ with refined particles near to fusion line; (**D**) weld with dendritic structure and eutectics in the inter-dendritic areas; visible porosity in HAZ and weld; (**E**) analysis of distribution of the most important elements in the weld metal (red rectangular indicate the EDS placement).

**Figure 17 materials-14-03124-f017:**
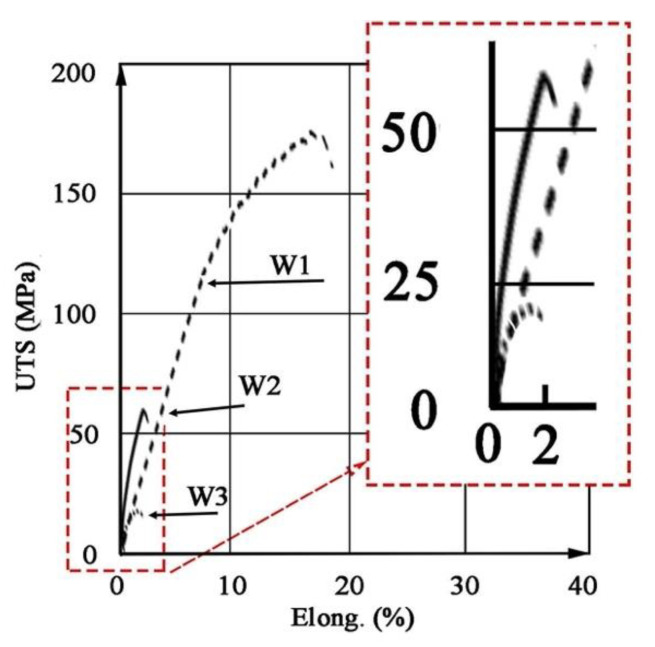
Stress–strain curves for the welded joins W1, W2, W3.

**Figure 18 materials-14-03124-f018:**
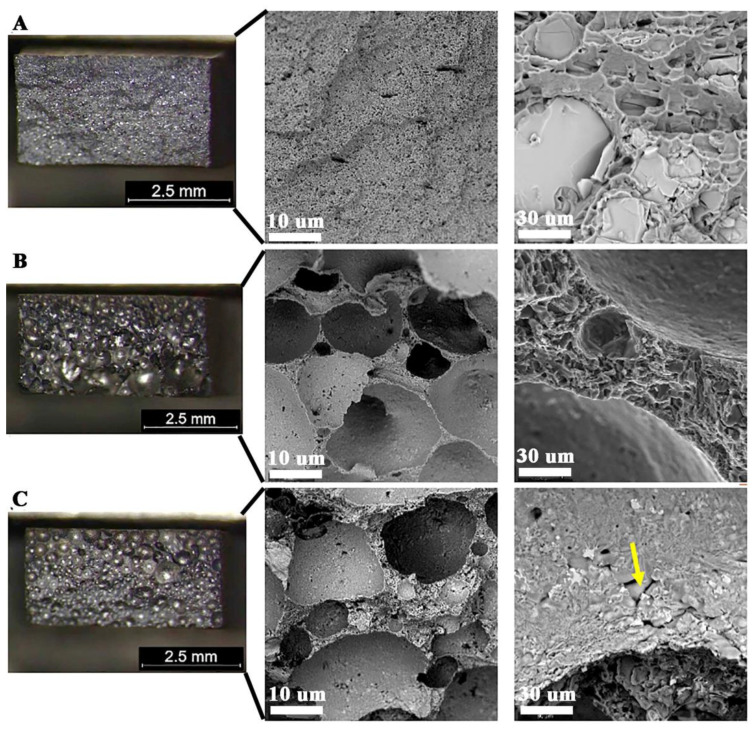
Quasi-ductile fracture of welded joints; visible porosity of W2 and W3; (**A**) W1—visible large Si precipitation, (**B**) W2, (**C**) W3.

**Figure 19 materials-14-03124-f019:**
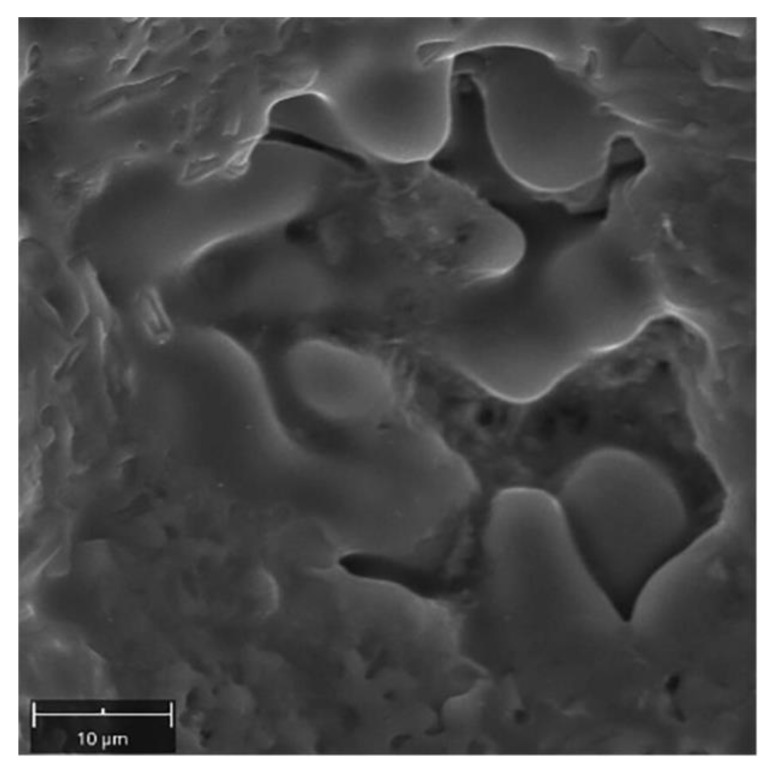
Surface of pore with visible effect of crystallization.

**Figure 20 materials-14-03124-f020:**
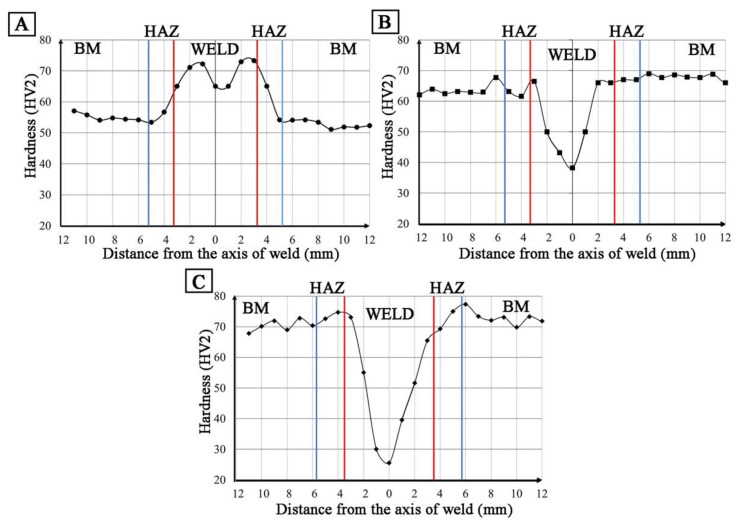
Hardness distribution in the cross-section of the welded joint: (**A**) W1, (**B**) W2, (**C**) W3.

**Table 1 materials-14-03124-t001:** Chemical composition and selected mechanical properties of AlSi11 alloy [31].

Element	Si	Fe	Cu	Mn	Mg	Zn	Ni	Ti
**Required by EN 1706 for Base Metal**	10–11.8	≤0.19	≤0.05	≤ 0.1	≤0.45	≤0.07	-	≤0.15
**Ingot AlSi11**	11.55	0.31	0.09	0.55	0.12	0.07	0.01	-
**Filler Metal AlSi12**	11-13	≤0.6	≤0.3	≤0.15	≤0.1	≤0.2	-	≤0.15

**Table 2 materials-14-03124-t002:** Mechanical properties of base metal AlSi11 and filler metal AlSi12 [31].

Properties	Ultimate Tensile Strength (UTS), MPa	Yield Stress (YS), (MPa)	Elongation (A), (%)	Hardness, (HB)	Density, (g/cm^3^)
**Base Metal Acc. to EN 1706**	≥150 (170)	≥70 (80)	≥6 (7)	≥50 (45)	2.65
**Filler Metal Acc. to Supplier**	≥130	≥60	≥5		

**Table 3 materials-14-03124-t003:** Samples numbering.

Sample	References Material	Large Chips	Fine Chips
**Base Materials**	M1	M2	M3
**Joints**	W1	W2	W3

**Table 4 materials-14-03124-t004:** Density measurements results consolidated after hot co-extrusion.

Sample	Density, (g/cm^3^)
**M1**	2.668
**M2**	2.667
**M3**	2.662

**Table 5 materials-14-03124-t005:** Average Vickers hardness (HV2) in cross-section of bars.

Sample	Hardness	St. Dev.
**M1**	55	1.77
**M2**	65	1.87
**M3**	74	2.51

**Table 6 materials-14-03124-t006:** Mechanical properties of extruded bars.

Sample	Ultimate Tensile Strength (UTS), (MPa)	Yield Stress (YS), (MPa)	Elongation (A), (%)
**M1**	167	120	18
**M2**	212	155	16
**M3**	224	160	12

**Table 7 materials-14-03124-t007:** Fraction of porosity in cross-section of the welds (with excess weld metal).

Sample	% of Porosity	Weld Cross-Section Area, (mm^2^)
**W1**	5.2 ± 0.3	21
**W2**	27.0 ± 0.3	39
**W3**	60.0 ± 0.3	50

**Table 8 materials-14-03124-t008:** Mechanical properties of the welded joints.

Sample	Ultimate Tensile Strength (UTS), (MPa)	Yield Stress (YS), (MPa)	Elongation (A), (%)
**W1**	173	130	6
**W2**	60	40	3
**W3**	20	15	3

## Data Availability

All data are provided in full in the results section of this paper.

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
