# Peer review of "Analysis of Microstructure and Mechanical Properties of AlSi11 after Chip Recycling, Co-Extrusion, and Arc Welding"

_materials, 2021, doi:10.3390/ma14113124_

Round 1

Reviewer 1 Report

The article is devoted to a relevant topic, namely the reuse of materials, resource conservation. The article is very well structured and well presented the results of the research. However, some improvement in the presentation of research results should be done. This will help the reader to better understand the presented studies.

1) It is necessary to bring a photo or drawing (sketch) of the laboratory installation at which extrusion was carried out.

2) Figure 2 shows photographs of machined aluminum alloy chips. Specify how the size of the chips and their average value were determined?

3) In Figures 7, 14 and 15, you need to sign the phases and inclusions. Their description is given in the text, but it will be more convenient for the reader if they are plotted in the figures.

Author Response

Thank you very much for taking your time to read our manuscript thoroughly and make valuable recommendations for its correction and improvement. Changes in the article are marked in yellow.

Reviewer 2 Report

The authors presented a very interesting topic from both a scientific and an industrial point of view. The paper is very well written, the results are accompanied by our own measurements. In my opinion, minor corrections are necessary before publication as follows:

  1. Some quantitative findings should be added to the Abstract.
  2. Introduction section - define more clearly the main objectives of the paper and indicate the novelties of the paper.
  3. Table 2, UTS and YS, do not use abbreviations but the full name (Ultimate Strength and Yield Stress?). The same for other tables.
  4. Conclusion section, indicate possible further directions of research, if any.

Author Response

Thank you very much for taking your time to read our manuscript thoroughly and make valuable recommendations for its correction and improvement. Changes in the article are marked in yellow

Reviewer 3 Report

The reviewer comments of the paper «Analysis of microstructure and mechanical properties of AlSi11 after chip recycling, co-extrusion and arc welding»

- Reviewer

The authors presented an article «Analysis of microstructure and mechanical properties of AlSi11 after chip recycling, co-extrusion and arc welding». However, there are several points in the article that require further explanation.

Comment 1:

Abstract.

Add qualitative and quantitative research findings. What is the novelty and practical value of the article?

Comment 2:

Introduction needs to be improved.

It is important, instead of group citing, to briefly describe the characteristics of each article. ... [...], ... [...]. What is their feature?

It is useful to review the articles:

Advances in Materials Science and Engineering, 2019, 2019, 4156176. DOI: 10.1155/2019/4156176

Metals 2018, 8, 394. DOI: 10.3390/met8060394

After the purpose of the article, briefly describe what has been done in each section.

Comment 3:

2 Experimental methodology

Tables 1 and 2 provide a citation [...]. Describe in more detail in Figures 1 and 5.

For devices and machine used in research, indicate in parentheses (manufacturer, city, country).

Comment 4:

  1. Test results

The resolution and quality of all figures needs to be improved.

Are all figures original? If not, then you need appropriate citations and publisher permissions.

Describe in more detail in Figures 6, 7, 8, 9, 12, 13, 14, 15, 16, 17, 18, 20, 21, 22. One figure - one paragraph description. The figure or table should be located immediately after the paragraph where it was described.

Figure 9 is best redrawn in color.

Sections 3 and 4 are best combined into one

  1. Results and discussion

As it stands, Section 4 refers to the figures and tables of Section 3, which is very inconvenient for the reader.

Comment 5:

It will be useful to add a section of Nomenclature in which to sign all the physical quantities and abbreviations encountered in the article. There are many physical quantities in the text and such a section will help to find the description of the necessary element.

For example,

FSW         : Friction stir welding

etc.

Comment 6:

Conclusions.

It is necessary to more clearly show the novelty of the article and the advantages of the proposed method. What is the difference from previous work in this area? Show practical relevance. What is the difference from other researchers? What are the quantitative and qualitative research results obtained?

The theme of article is interesting. Authors should carefully study the comments and make improvements to the article step by step. Highlight all changes. After major changes can an article be considered for publication in the "Materials".

Author Response

(The authors gave the same response as above.)

Round 2

Reviewer 3 Report

The authors have improved the article according to the comments. The article can be accepted for publication. However, before accepting the article for the authors, it is important: specify the composition of the authors for article 36.